# Analyzing Lockdown Policies and Their Effectiveness in Romania and Hungary

**Stefania Kerekes** [1,2], **Ariadna Georgiana-Eugenia Badea** [3] **and Dragos Paun** [4,*]

1   Faculty of European Studies, Babeș-Bolyai University, 400090 Cluj-Napoca, Romania;
    stefaniakerekes@gmail.com
2   Department of Epidemiology, School of Public Health, University of Michigan, Ann Arbor, MI 48109, USA
3   "Iuliu Hațieganul" University of Medicine and Farmacy, 400012 Cluj-Napoca, Romania;
    badeaariadna@gmail.com
4   Faculty of Business, Babeș-Bolyai University, 400174 Cluj-Napoca, Romania
*   Correspondence: dragos.paun@ubbcluj.ro; Tel.: +40-264-599-170

**Abstract:** There has been a debate on the efficiency of lockdown policies worldwide, and several researchers have studied this aspect by trying to implement different statistical models. The aim of the research was to compare two countries with similar lockdown policies and observe the impact of the total lockdown policy on the spread of the COVID-19 disease. Taking in consideration that the total lockdown in Romania lasted for 52 days and in Hungary for 54 days, we would like to see how the infection rate changed with every week of the lockdown by obtaining an average for every week (7 days) divided by the total lockdown days in each country. The values that we took in consideration are as follows: the daily infected cases, the daily infected cases/million, the daily cases of death and the daily cases of death/million in both countries. We tried to apply the same rule after the end of the total lockdown and observe the outcomes. The results showed that the minimum number of days to observe the effects of total lockdown and the effects after the lockdown was 21 (3 weeks) in both countries.

**Keywords:** coronavirus; COVID-19; health policy; lockdown





## 1. Introduction

The virus called severe acute respiratory syndrome coronavirus 2 (SARS-CoV-2) started to spread in Wuhan, China, in December 2019 and emerged rapidly around the globe, causing millions of deaths [1–4]. In response, the World Health Organization (WHO) officially declared a "global pandemic" on 11 March 2020 [5]. Admitting the international crisis, the authorities were forced to impose rigorous measures on the general population, for example, total lockdown policies, school closures, social distancing, border closures, etc., which led to a global economic collapse. The lack of standardized guidelines, differences between countries and their health policies led to numerous deaths [6]. Even if non-pharmaceutical measures were implemented, international collaboration was needed for managing the global crisis by elaborating new guidelines, adjusting existing strategies and developing vaccines [7].

Due to the globalized world that involves a rapidly growing and mobile population, different urbanization trends, increased food consumption and developed global transportation, the outbreak of a pandemic was inevitable. All of these aspects contributed to the spread of the pathogen [8–10]. Since we cannot stop the cross-border flow of people, universal vaccination is needed to prevent future outbreaks [11].

Several researchers studied the effectiveness of lockdown measures over time. Carefully consulting the literature, we found that countries implemented lockdown policies in different periods of the year 2020. The countries adopting lockdown measures earlier, could control the pandemic better than those whom adopted the policy later [12]. Thus,

the benefits of lockdown policies could be observed in time [13]. The countries that implemented lockdowns, showed a great decrease in the number of new cases of COVID-19 in comparison with the countries where a lockdown was not an option [13]. Furthermore, the number of lockdown days differed in each country, and this fact could explain the big differences in the numbers of newly infected cases and the differences in reaching the peak in the year of 2020 [12].

Moreover, lockdown policies have to be implemented very carefully, because of negative effect on the economy and on people's mental health. Earlier lockdowns might save several lives, but dealing with the economic consequences in the end might be difficult. Later lockdown results in a smaller financial loss but a greater human loss. Therefore, the decision of lockdown enforcement has to be carefully considered [12–14].

Despite of several negative effects, lockdown policies had their positive results, too, like the renewal of the environment, which could be observed in increased air and water quality [12]. Researchers pointed out that lifting all of the restrictions suddenly, would have had dramatic implications, like the immediate increase in infections. This is the reason why restrictions had to be lifted progressively [13]. Other studies demonstrated that populations with a large number of elderly (e.g., Italy) are more likely to report a higher death rate. Similarly, people living closely to each other (e.g., in big cities) are more susceptible for contacting the virus [14,15]. Thus, total lockdown policies are effective in locations with high infection rates [12].

In conclusion, the reason for choosing Romania and Hungary in particular was because of the similarities in applied lockdown policies, the proximity of the countries and the common ideology from the past. We tried to find answers for total lockdown policy effectiveness by analyzing European countries with similar lockdown policies and by consulting the previous literature on this subject. Our final goal was to determine if total lockdown policies are really necessary for controlling a pandemic like COVID-19.

## 2. Materials and Methods

Our datasets were from the GitHub website [16], which contains a collection of data from the Johns Hopkins University online COVID-19 interactive dashboard [17].

The study analyzed two timeframes in both countries, during the lockdown and after the lockdown, considering the fact that people respected the policy in both countries. We emphasized eight weeks during the lockdown in 2020 and eight weeks after the lockdown in 2020 in both countries. More precisely, in Romania, we took the following timeframes: between 24 March and 15 May (lockdown, 52 days) and between 15 May and 10 July 2020 (after the lockdown). In Hungary, we analyzed the following timeframe: between 11 March and 4 May (54 days) and between 4 May and 29 June (after the lockdown). We selected these two countries because of their proximity and due to the similarity of the two healthcare systems. Both countries have a system which is centralized (funded by the government) but also features a lot of development in private healthcare as well. Hungary is established as a hub for medical tourism with regard to dentistry and Romania is on its way to becoming such a hub. This is the reason for a large number of people living abroad which still have ties to the respective countries and also to the performance of the medical system coupled with the lower cost compared to western European countries to come. With regard to the health of the population, we see that in both countries, we have an aging population, but there is a difference in the rural population—in Romania, we have 45% of the population living in rural areas, while Hungary is at 28%. This factor could be considered important for the COVID-19 outbreak because we have seen that the coronavirus is transmitted within dense populations.

To pursue our objectives, we organized our dataset in tables with the average values of seven-day cases divided by the number of lockdown days in each country (Tables 1 and 2). We applied the same method with the values after the lockdown and organized them in tables (Tables 3 and 4).

**Table 1.** Average number of cases during the lockdown in Romania.

| Weeks | Days | Average Daily New Confirmed Cases | Average Daily New Confirmed Deaths | Average Daily New Confirmed Cases/1,000,000 | Average Daily New Confirmed Deaths/1,000,000 |
|---|---|---|---|---|---|
| 1 | 52 | 298.6 | 21.0 | 15.5 | 1.1 |
| 2 | 52 | 281.6 | 20.4 | 14.6 | 1.1 |
| 3 | 52 | 256.7 | 18.7 | 13.3 | 1.0 |
| 4 | 52 | 234.6 | 17.6 | 12.2 | 0.9 |
| 5 | 52 | 227.2 | 16.3 | 11.8 | 0.8 |
| 6 | 52 | 228.7 | 14.8 | 11.9 | 0.8 |
| 7 | 52 | 233.8 | 13.9 | 12.1 | 0.7 |
| 8 | 52 | 262.6 | 13.9 | 13.7 | 0.7 |

**Table 2.** Average number of cases during the lockdown in Hungary.

| Weeks | Days | Average Daily New Confirmed Cases | Average New Daily Confirmed Deaths | Average Daily New Confirmed Cases/1,000,000 | Average Daily New Confirmed Deaths/1,000,000 |
|---|---|---|---|---|---|
| 1 | 54 | 58.8 | 7.6 | 6.1 | 0.8 |
| 2 | 54 | 60.9 | 8.3 | 6.3 | 0.9 |
| 3 | 54 | 59.3 | 8.6 | 6.1 | 0.9 |
| 4 | 54 | 55.6 | 8.7 | 5.8 | 0.9 |
| 5 | 54 | 44.7 | 7.6 | 4.6 | 0.8 |
| 6 | 54 | 36.0 | 6.4 | 3.7 | 0.7 |
| 7 | 54 | 26.4 | 5.1 | 2.7 | 0.5 |
| 8 | 54 | 19.6 | 4.0 | 2.0 | 0.4 |

**Table 3.** Average number of cases after lockdown in Romania.

| Weeks | Days | Average Daily New Confirmed Cases | Average Daily New Confirmed Deaths | Average Daily New Confirmed Cases/1,000,000 | Average Daily New Confirmed Deaths/1,000,000 |
|---|---|---|---|---|---|
| 1 | 52 | 244.7 | 13.8 | 12.7 | 0.7 |
| 2 | 52 | 285.8 | 14.0 | 14.9 | 0.7 |
| 3 | 52 | 355.9 | 15.2 | 18.5 | 0.8 |
| 4 | 52 | 474.1 | 17.0 | 24.6 | 0.9 |
| 5 | 52 | 608.3 | 19.9 | 31.6 | 1.0 |
| 6 | 52 | 736.0 | 23.6 | 38.3 | 1.2 |
| 7 | 52 | 861.0 | 27.1 | 44.8 | 1.4 |
| 8 | 52 | 966.6 | 30.3 | 50.2 | 1.6 |

**Table 4.** Average number of cases after the lockdown in Hungary.

| Weeks | Days | Average Daily New Confirmed Cases | Average Daily New Confirmed Deaths | Average Daily New Confirmed Cases/1,000,000 | Average Daily New Confirmed Deaths/1,000,000 |
|---|---|---|---|---|---|
| 1 | 54 | 20.5 | 4.3 | 2.1 | 0.4 |
| 2 | 54 | 16.5 | 3.2 | 1.7 | 0.3 |
| 3 | 54 | 13.0 | 2.6 | 1.3 | 0.3 |
| 4 | 54 | 10.0 | 2.0 | 1.0 | 0.2 |
| 5 | 54 | 10.1 | 1.3 | 1.0 | 0.1 |
| 6 | 54 | 9.7 | 1.0 | 1.0 | 0.1 |
| 7 | 54 | 10.1 | 0.8 | 1.0 | 0.1 |
| 8 | 54 | 13.8 | 0.7 | 1.4 | 0.1 |

Continuing our analysis, we obtained charts and determined the week with the most significant drop in infections for both countries. After consulting the literature and observing our results, we came to several conclusions presented below.

### 3. Results

Analyzing the results, we discovered that the most significant drop in cases was observed after three weeks (21 days) of the lockdown in both countries. Even though in Hungary the number of cases continued to increase in the first weeks of the lockdown, after the three-week period, there was definitely a decrease in infection rates.

The effects of lockdown policies are clearly visible in the year 2020, where the COVID-19 vaccine was not yet available. The values suggested that after the lockdown implementation, the effects of the policy could have been observed after a certain amount of time (see Figure 1). This time is considered to be three weeks (21 days). Therefore, spending the minimum time in lockdown and lifting restrictions progressively, low infection rates can be maintained.

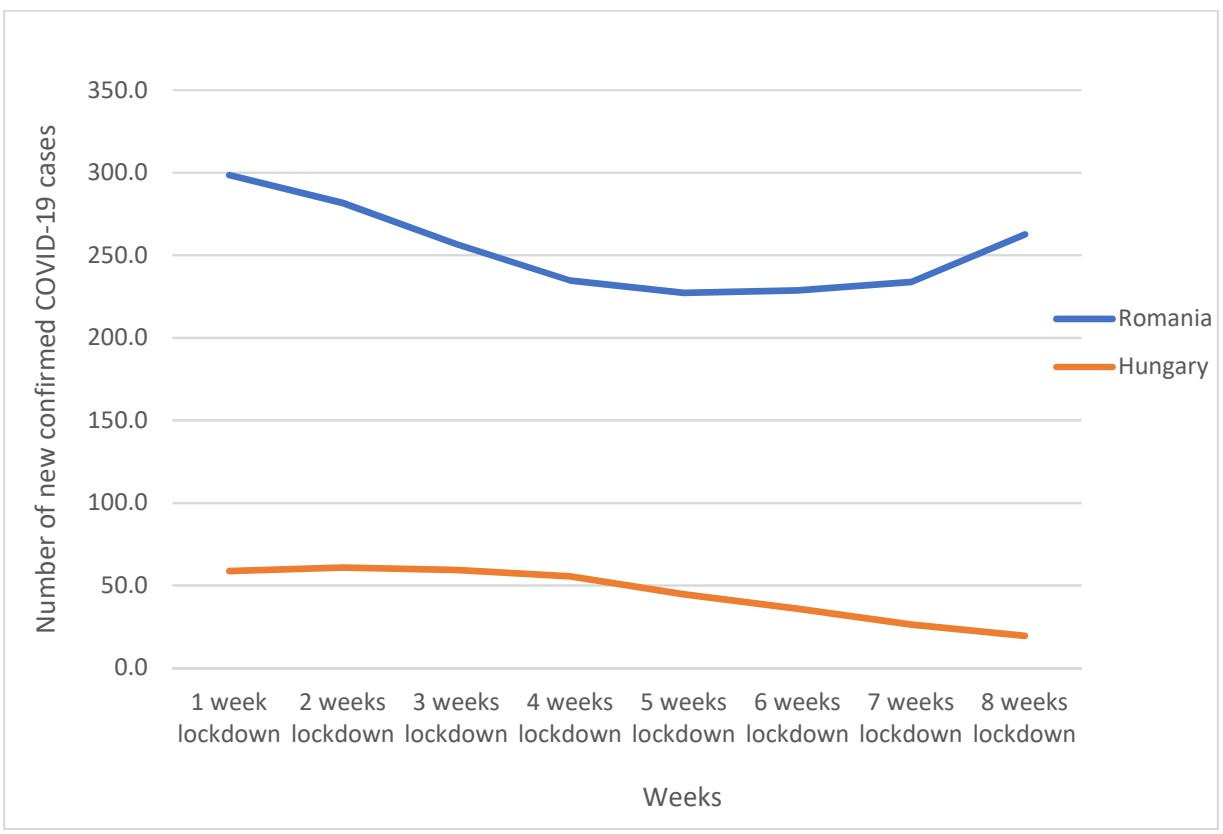

**Figure 1.** New confirmed COVID-19 cases.

The most significant drop in the average number of confirmed deaths was observed in Hungary compared with Romania, where we remarked a gradual drop (see Figure 2). Moreover, the average number of new confirmed cases/million decreased faster in Romania then in Hungary, where the drop was moderate (see Figure 3). This might suggest that the rules of lockdown policies were followed more rigorously in Romania then in Hungary.

Regarding the charts, the most spectacular results could be observed in the average number of new deaths/million (see Figure 4). In Hungary, after three weeks of the lockdown, the average number of death cases started to drop significantly, and in Romania, we could observe only a gradual reduction.

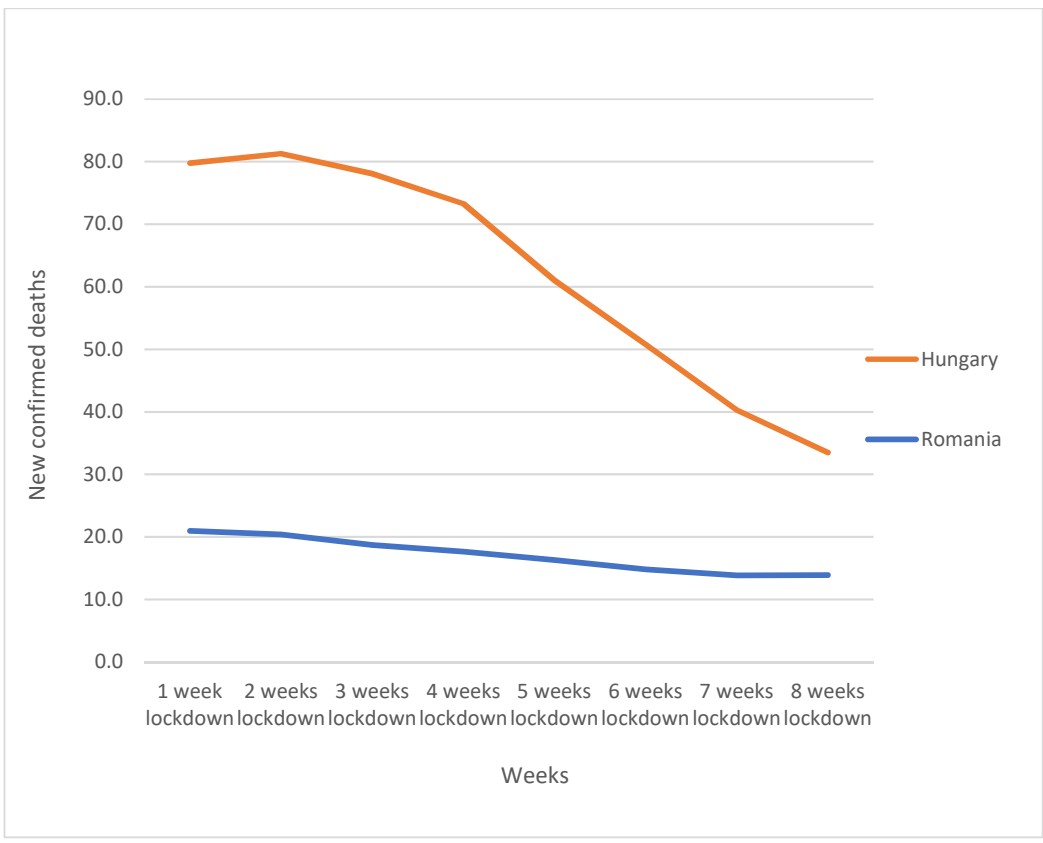

**Figure 2.** New confirmed COVID-19 deaths.

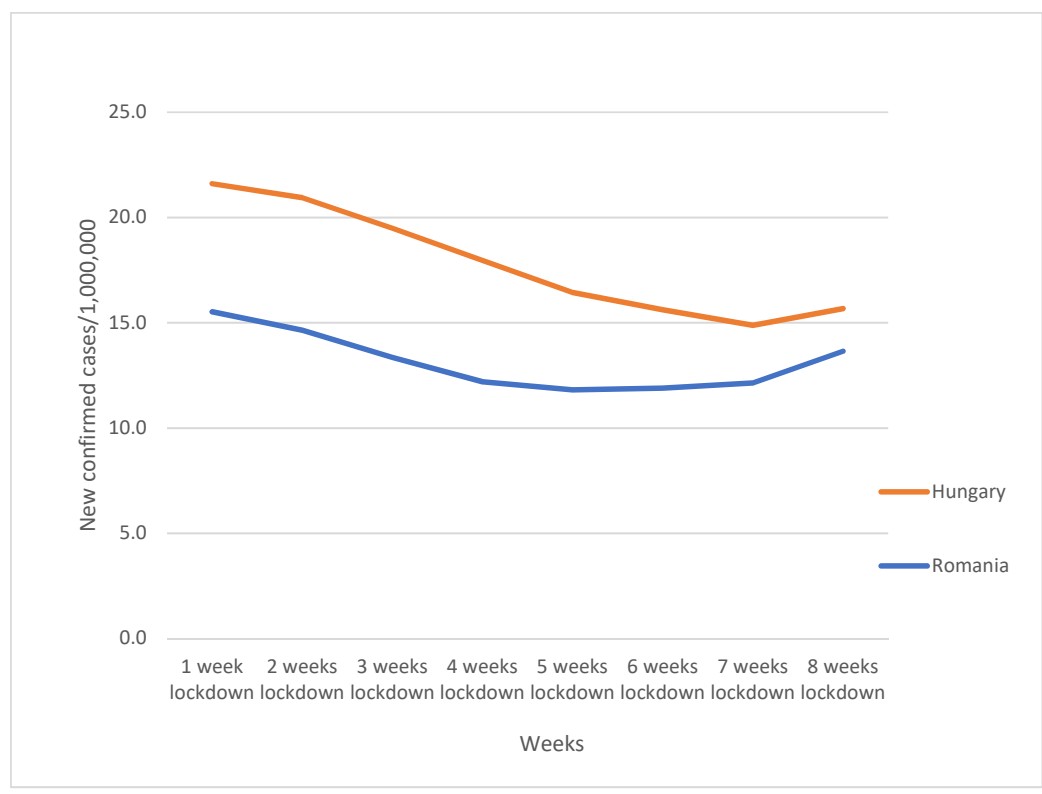

**Figure 3.** New confirmed cases/1,000,000.

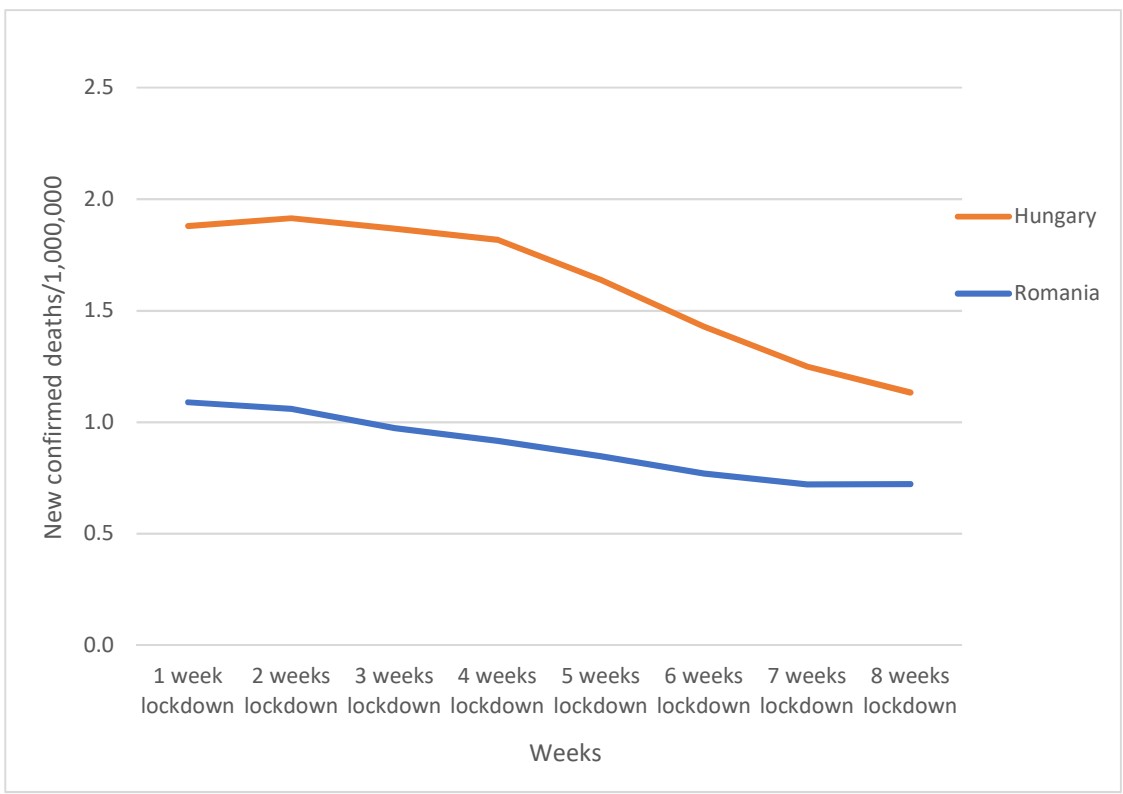

**Figure 4.** New confirmed COVID-19 deaths/1,000,000.

To take the research further, we analyzed what happened every week after the lockdown ended with the same method of analysis. The period after the lockdown is quite interesting because the restrictions are lifted and the average number of infected cases increases day by day. Looking at the average number of cases/million, the values did not change that much in time. This shows that lifting restrictions progressively is highly important for the control of the pandemic.

In conclusion, according to the result, lockdown policies were visibly effective in both countries at a time when vaccination strategies were not yet implemented. This type of a policy is useful to avoid hospital overcrowding and slow the spread of the disease until a vaccine is discovered. Total lockdown policies are highly efficient in urban areas with a dense population and can clearly save lives if the policy is implemented in a timely manner.

## 4. Discussion

The findings from several research articles suggest that low level of national preparedness, high GDP, advanced age and the presence of other medical conditions are strongly connected with high mortality rate and increased number of infections because of COVID-19. It is clear that early detection, international cooperation and a stable healthcare system are crucial in stopping the pandemic [2]. A great level of national preparedness is reflected in timely implementation of health policies, appropriate public health guidance, optimal population compliance and so on. A more restrictive national health policy, such as lockdown measures are associated with less disease transmission [2].

Lockdown policies can be divided in two parts. The first part contains the decision to implement lockdown measures at a certain epidemiological time and the second part is the level of severity of the lockdown. Some countries managed to control the pandemic without implementing a formal lockdown during the first wave of 2020, and the key to a successful strategy was the discipline of the people and the number of tests taken [18].

During the first wave in 2020, only a few countries decided to implement early and moderate response strategies, and Hungary was one of them. Romania also implemented

the decision to enter a lockdown during the first wave, early in epidemiological time, which meant that the pandemic was not out of control yet, and this was the reason why the country had a relatively low peak in the first wave [18].

Introducing severe lockdown policy from the beginning could be a key to success in pandemics, but there is always the risk that the virus might re-entre in the country in any moment which means new infections of COVID-19. Lockdown policies are convenient at the beginning of an outbreak, but on long term might come with more harm, then good. Finding the most suitable measures which will decrease the rate of infections and keep the functionality of the economy would be plausible in the future [19].

## 5. Limitations of the Study

The pandemic is still ongoing, and our results did not cover all the European countries to better compare lockdown policies. Moreover, to conduct a deeper analysis, several details have to be taken in consideration, like GDP, age category, presence or absence of chronic diseases, ethnicity, etc. Hopefully, in the future, this study will be developed further.

Moving forward, our research comes as a support for future policymakers in deciding which health policy is suitable for a future pandemic to be controlled. In addition, the negative effects of total lockdowns are a different study subject that could be analyzed in future studies to have a complete picture of health policies.

## 6. Conclusions

The pandemic caused by SARS-CoV-2 took the world by surprise. Global economies have collapsed, mental health has been affected, medical systems have been overwhelmed by the numbers of hospitalizations and deaths. Our study, although conducted in only two countries and over a period of 8 weeks, reiterates the importance of research in this area to come up with practical and realistic solutions to help governments manage global issues such as the current pandemic. Given the unlimited mobility of the population, such a pandemic may occur again, and then it is very important to know, as our study shows, what steps must be followed both to control an outbreak and not to destabilize the economic environment, health, etc. Thus, our study is of major importance because it clearly shows the difference between the number of cases of infection during the total lockdown period and during the gradual relaxation period so we can learn and take further actions from the conclusions.

**Author Contributions:** Conceptualization, S.K.; methodology, S.K.; software, S.K.; validation, S.K. and A.G.-E.B.; formal analysis, S.K.; investigation, S.K.; resources, S.K.; data curation, S.K.; writing—original draft preparation, S.K. and A.G.-E.B.; writing—review and editing, D.P.; visualization, D.P.; supervision, D.P.; project administration, S.K. and A.G.-E.B.; funding acquisition, D.P. All authors have read and agreed to the published version of the manuscript.

**Funding:** This work was supported by a grant of the Romanian Ministry of Education and Research, CNCS-UEFISCDI, project number PN-III-P4-ID-PCE-2020-2174, within PNCDI III. The publication of this article was partially supported by the 2020 Development Fund and other funds of the Babeş-Bolyai University.

**Institutional Review Board Statement:** Not applicable.

**Informed Consent Statement:** Not applicable.

**Acknowledgments:** The authors thank to Abram L. Wagner from the Department of Epidemiology, University of Michigan, USA, for interesting comments and suggestions.

**Conflicts of Interest:** The authors declare no conflict of interest.

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
