# Peer review of "Analyzing Lockdown Policies and Their Effectiveness in Romania and Hungary"

_challenges, doi:10.3390/challe12020020_

Round 1

Reviewer 1 Report

The paper is interesting, but it must be improved in relation to methodology and presentation of results for accurate conclusions.

Some suggestions:

Introduction

Correct the reference [14]; it appears as {Balmford, 2020 #8].

Methodology

Some nformations are necessaries to understand the results:

Inform the time for data analyzed. Which period do data refer to?

Did People join the lockdown for both countries?

It is important to know if a drop of infection in Hungary is effect of lockdown or/and of vaccination. I think the vaccination has been started in Hungary first, in relation to Romania.

The days of lockdown cannot be used to obtain a mean of infected because this number is different between countries compared to each other. Moreover, for comparison effect, is preferable cases/million instead of absolute number.

Results from the last week cannot be compared with ones from previous weeks because the number of days is different (last week has no 7 days).

Results

Keep from 1 to 8 in column "Weeks", in Tables 3 and 4, as Tables 1 and 2.

Show results in bars instead tables; in the same graphic, show infection during and after lockdown to facilitate visualization and conclusion.

Data presented in terms of percentage are more interesting and easier to analyze. A variation of infection between weeks, in terms of percentage, could be interesting.

Authors affirm "Looking at the tables, we observed that after lockdown implementation, the numbers started to drop every week." However, it is valid to Hungary, only. The cases increase in Romania after lockdown. See:

Romania: new cases during lockdown (298,281,256,234,227), against after lockdown (244, 285, 355, 474, 608).

Hungary: new cases during lockdown (58,60,59,55,44), against after lockdown (20,16,13,10,10).

Consider a statistical analysis between data during and after lockdown, as well as to compare data between countries.

Author Response

First of all we want to thank the reviewer for the suggestions. We believe that these will improve our paper. We will answer point by point the comments addressed by the reviewer. 

Correct the reference [14]; it appears as {Balmford, 2020 #8].

- We have corrected the reference

Methodology

Some nformations are necessaries to understand the results:

Inform the time for data analyzed. Which period do data refer to? Did People join the lockdown for both countries?

Response: Indeed the people reacted to the lockdown measures – and this information was included in lines 101-108.

It is important to know if a drop of infection in Hungary is effect of lockdown or/and of vaccination. I think the vaccination has been started in Hungary first, in relation to Romania.

Response: The impact of the vaccination was not the topic of the current research. We thank the reviewer for opening this interesting idea for research and comparison. Our hypothesis stands that the effects that we see are coming from the lockdown measures.

The days of lockdown cannot be used to obtain a mean of infected because this number is different between countries compared to each other. Moreover, for comparison effect, is preferable cases/million instead of absolute number.

Response: 

Indeed we have seen that there is a difference between the lockdown measures. However, in our analysis we have included a daily average of new cases during those weeks. At your suggestion we have changed the header and included daily. We have also calculated in the last column the average cases per million of population.

Results from the last week cannot be compared with ones from previous weeks because the number of days is different (last week has no 7 days).

Response: 

Indeed the lockdown measures were not totaling full weeks, but in our analysis we have worked with average daily rates so that we have divided the total number by 7 in the full weeks and the weeks that were not containing 7 days we have divided by the number of days in that week, thus having am average which is comparable.

Results

Keep from 1 to 8 in column "Weeks", in Tables 3 and 4, as Tables 1 and 2.

Show results in bars instead tables; in the same graphic, show infection during and after lockdown to facilitate visualization and conclusion.

Response: Thank you very much, we have changed the tables, it was a typo.

Data presented in terms of percentage are more interesting and easier to analyze. A variation of infection between weeks, in terms of percentage, could be interesting.

Response: 

We have included in our first draft a visualization in form of bars but the results were not so conclusive. With your permission we would like to keep the current visualization with numbers as we consider that it makes a better comparison. 

Authors affirm "Looking at the tables, we observed that after lockdown implementation, the numbers started to drop every week." However, it is valid to Hungary, only. The cases increase in Romania after lockdown. See:

As the two countries have a considerable variation of population a calculation based on percentage would not give the perfect image. More so, in the official reports we have seen that cases / million or thousand of inhabitants is used as a refence and we have kept this way. With your permission we would like to keep the same format. The average cases were also selected because of comparison as the lockdown days were different (52 compared to 54).

Romania: new cases during lockdown (298,281,256,234,227), against after lockdown (244, 285, 355, 474, 608). Hungary: new cases during lockdown (58,60,59,55,44), against after lockdown (20,16,13,10,10). Consider a statistical analysis between data during and after lockdown, as well as to compare data between countries.

We thank the reviewers for this information. We have mentioned in the articles that a change is visible after 14-21 days (see line 92-99) . However due to the nature of the specific language, we the authors wanted to say that the number of cases decreased after the implementation of the lockdown => see table 1 (from 298.9 => 228). After the lockdown was lifted, the number of cases increased just like as the distinguished reviewer pointed out. At the suggestion of the reviewer we have submitted the article to a proofreading service to make sure that these errors are avoided. 

Reviewer 2 Report

The authors made a good attempt to analyse the COVID cases and deaths in Hungary and Romania and examined the role of lockdown in the reduction in infection and deaths. 

I have few comments to improve the paper:

  1. Authors need to provide more information on lockdown exact months and years in the paper. As it is not very clear.
  2. Authors need to provide more background of both the countries in general in the introduction, mainly to strengthen their case to study these specific countries only. Some statistics of their health condition, infrastructure, population, and demographic characteristics can be provided. 
  3. Authors need to write their key findings in the discussion part and correlate their discussion around those findings. The authors need to explain more clearly the last paragraph starting with "Finding the most suitable health policy". How it's related to the current study is not clear. 
  4. There is a need to recheck all the figures, as some discrepancy in the labels and format is observed. 
  5. There is a need to rewrite few sentences in the text. I will suggest proofreading by a native English speaker. 
  6. Please also check the style of reference where multiple references are listed in the same place. It should follow this journal referencing style.  

Author Response

First of all we would like to thank the reviewer for the suggestions. We think that these would improve the quality of the paper and we appreciate the time to review the paper. We will answer each comment:

I have few comments to improve the paper:

1. Authors need to provide more information on lockdown exact months and years in the paper. As it is not very clear.

Response: Thank you very much we have included this information in the lines 79-83.

2. Authors need to provide more background of both the countries in general in the introduction, mainly to strengthen their case to study these specific countries only. Some statistics of their health condition, infrastructure, population, and demographic characteristics can be provided. 

Thank you very much, we have included some extra information which is specific to the paper in lines 84-95.

3. Authors need to write their key findings in the discussion part and correlate their discussion around those findings. The authors need to explain more clearly the last paragraph starting with "Finding the most suitable health policy". How it's related to the current study is not clear. 

Thank you very much we have reviewed this section and hope that it is clearer now. 

4. There is a need to recheck all the figures, as some discrepancy in the labels and format is observed. 

Response: This suggestion was made by another reviewer also. We apologize for this error and we have corrected now the figures and tables. 

5. There is a need to rewrite few sentences in the text. I will suggest proofreading by a native English speaker. 

Response: We have sent the paper to a specialised person. It might happen that when you are in the text you oversee the errors. Thank you for this suggestion.

6. Please also check the style of reference where multiple references are listed in the same place. It should follow this journal referencing style.

Reponse: We thank the reviewer for this comment. We have used the MDPI reference style which is provided and used Endnote X9 to format the references. 

Round 2

Reviewer 1 Report

That is a very interesting/important contribution! Regards,

Reviewer 2 Report

Thank you for incorporating all the changes proposed earlier. I have no further comments. 

This manuscript is a resubmission of an earlier submission. The following is a list of the peer review reports and author responses from that submission.